# Uptake of COVID-19 vaccine and associated factors among Health Care Workers at Queen Elizabeth Central Hospital in Malawi: A cross-section study

**Angella Joseph Chikhoza**[1]*, **Wingston Ng'ambi**[1], **Alinane Linda Nyondo-Mipando**[1,2]

**1** Department of Health Systems and Policy, School of Global and Public Health, Kamuzu University of Health Sciences, Blantyre, Malawi, **2** Department of Women's and Children's Health, University of Liverpool, Liverpool, United Kingdom

* angechikhoza@gmail.com

## Abstract

Globally, the introduction of the Coronavirus Disease of 2019 (COVID-19) vaccine within one year of the pandemic brought doubts to the general population including health professionals. Even though Health Care Workers (HCWs) are at a high risk of contracting and transmitting the coronavirus due to their work, their COVID-19 vaccine uptake is unknown. This study aimed to determine the uptake of the COVID-19 vaccine and identify factors associated with uptake among HCWs at Queen Elizabeth Central Hospital (QECH) in Blantyre, Malawi. We conducted a quantitative cross-sectional study at QECH. HCWs over 18 years, and with at least one year working experience at QECH were recruited using a simple random sampling technique between December 2021 and February 2022. Data were collected on sociodemographic characteristics, medical history, COVID-19 and COVID-19 vaccine perspectives. The study was informed by Behavioral and Social Drivers of COVID-19 Vaccination framework. We computed vaccine uptake as number of HCWs fully vaccinated divided by total number of participants recruited in the study. Bivariate and multivariable logistic regression models were used to identify factors associated with vaccine uptake. Of the 273 HCWs enrolled into the study 71% were fully vaccinated. Gender (Adjusted Odds Ratio (AOR) = 0.54, 95% CI: 0.30–0.96), ownership of medical insurance schemes (AOR = 2.90, 95% CI: 1.36–6.18) and history of vaccine refusal (AOR = 0.29, 95% CI: 0.12–0.70) were significantly related to vaccine uptake. Age and work experience were statistically significant. The level of education, the income of HCWs and occupation did not determine whether HCWs got vaccinated against COVID-19. There was high vaccine uptake among HCWs at QECH. Some risk factors for severe COVID-19 such as advancing in age and work experience are persuasive to COVID-19 vaccination. We recommend intensified COVID-19 vaccination campaigns for subgroups such as young adults and female HCWs.

**Data Availability Statement:** The datasets used and/or analysed during the current study are all included in the manuscript as part of the results.

**Funding:** The authors received no specific funding for this work.

**Competing interests:** The authors have declared that no competing interests exist.

## Introduction

The severe acute respiratory syndrome coronavirus 2 (SARS-CoV-2) is in a group of enveloped, positive-sense, and single-stranded Ribonucleic Acid (RNA) viruses [1]. In humans, it is transmitted through droplets and close contact with the infected person [1]. The virus causes the Coronavirus Disease of 2019 (COVID-19) [1]. COVID-19 is causing high mortality and morbidity rates globally. As of 13 April 2024, 704,753,890 had been infected with coronavirus and 7,010,681 died [2]. HCWs such as doctors, clinicians, and nurses are on the frontline in fighting against the COVID-19 pandemic and remain the most affected population due to the nature of their work. In Malawi, according to Health Management Information System (HMIS) at Blantyre District Health Office (DHO), 1061 HCWs in the district tested SARS-CoV-2 antibodies positive as of August 2023.

Vaccination against SARS-CoV-2 contributed to the end of COVID-19 pandemic [3]. In December 2020, the first COVID-19 vaccines were authorized for emergency use and HCWs were among the priority groups to be vaccinated [3, 4]. Despite that COVID-19 vaccines were proven to be safe and effective in randomized clinical trials and real-world studies; delay and refusal of COVID-19 vaccine has been observed in the general population even among health professionals [3–5]. A high proportion of unvaccinated HCWs against COVID-19 can be a threat to themselves, patients, their families, and the general population [6]. In addition, the uncontrolled rising of COVID-19 among HCWs can paralyse the country's health care system [6].

Vaccine hesitancy is defined as the delay in acceptance, reluctance, or refusal of vaccination despite the availability of vaccination services [7]. COVID-19 vaccine hesitancy is still a major challenge worldwide [8]. In Malawi, AstraZeneca COVID-19 vaccines expired after the vaccines had only been in the country for 2 months due to slow roll out and uptake. However, despite the COVID-19 vaccine stock-outs and logistical issues, several factors influence its uptake among the HCWs [9]. In the United States of America (USA), a study found that the vaccine uptake among HCWs was associated with some social-economic factors like age, gender, and workplace. Earlier studies focused mainly on COVID-19 vaccine perceptions, acceptability, and hesitancy among HCWs [10]. However, there was limited information on the actual uptake of the COVID-19 vaccine and associated factors at tertiary hospitals in Malawi. Therefore, this study assessed the uptake of the COVID-19 vaccine and associated factors among HCWs at Queen Elizabeth Central Hospital (QECH) in Blantyre, Malawi. The study was informed by the WHO Behavioral and Social Drivers (BeSD) Framework for COVID-19 vaccination, which is presented in Fig 1 [11].

### Theoretical framework: WHO Behavioral and Social Drivers (BeSD) framework for COVID-19 vaccination

The framework was developed by the WHO working group experts in 2018 [11]. It focuses on the COVID-19 vaccine uptake for older adults and HCWs [12]. It explains the root causes of low vaccine uptake or stagnation [12]. BeSD of vaccination entails that measuring the behavioural and social drivers may determine peoples' intentions to get vaccinated [11]. The framework constitutes four main domains which include: what people think and feel about the vaccine, social processes that promote or hinder vaccination, what motivates individuals to get vaccinated, and the practical issues that influence people to seek and receive vaccination [11]. However, this study used variables that are coherent with the study objectives. It used domains from what people think and feel, and social processes. Other variables were excluded such as those from practical issues. The following diagram illustrates the BeSD of vaccination framework.

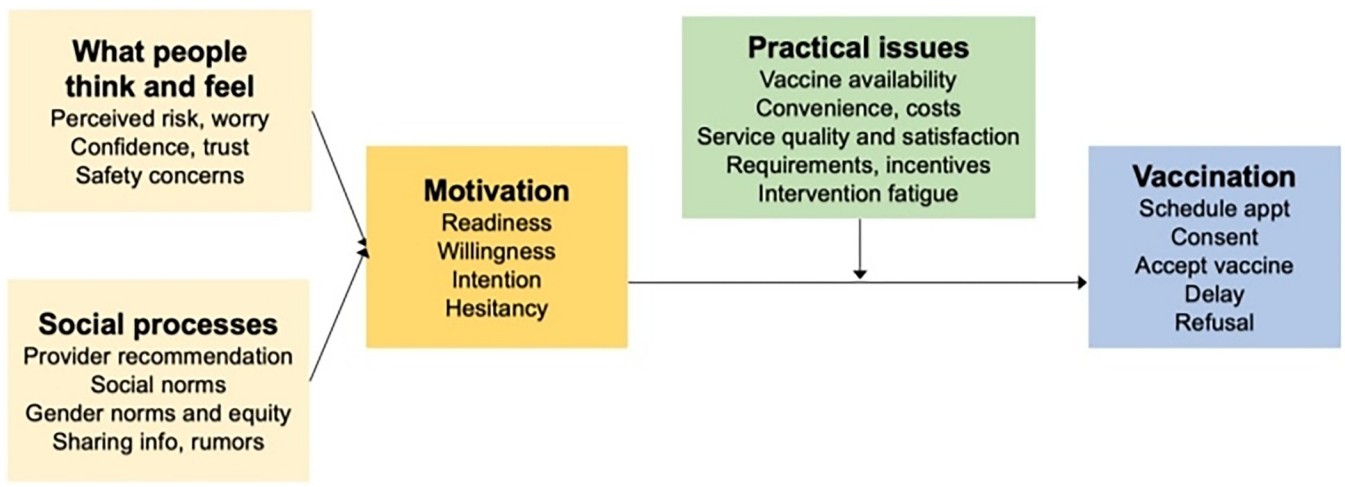

**Fig 1. The BeSD of vaccination framework [11].**

## Methods

### Study design

We conducted a quantitative, descriptive cross-section from November 2021 to August 2022. It was an appropriate study design since the study aimed at determining the vaccine uptake (outcome) and identifying associated factors (independent variables) within a short duration. Data were collected using a structured questionnaire.

### Study setting

The study was done at Queen Elizabeth Central Hospital (QECH) in Blantyre urban, Malawi. It is a 1250-bed public tertiary hospital under the Ministry of Health (MoH), located in the southern region of Malawi. The hospital has about 1,710 HCWs. We selected QECH because it is the largest tertiary level facility in the southern region of Malawi and has a catchment area of approximately six million people. The facility provides specialized services such as medical, surgical, orthopedics, obstetrics, gynecological, and pediatric. The HCWs at the facility consist of doctors, clinical officers, nurses, radiologists, pharmacists, physiotherapists, and laboratory technologists. There are various research groups and partners that operate and assist QECH in conducting research and health service delivery respectively, such as the Blantyre Malaria Project (BMP), Malaria Alert Centre (MAC), John Hopkins Malawi, International Training and Education Centre for Health (I-TECH) and the Malawi Liverpool Wellcome Trust (MLW).

The study participants were recruited from the following wards/department: the COVID-19 isolation units, ENT, HDRU, Male and Female Medical (3B and 4A), Male and Female Surgical (5A and 5B), Accidents and Emergency (A and E), AETC, Orthopaedic, Gynaecological, Chatinkha Nursery, Paediatric Moyo, Paediatric Special Care Ward (PSCW), Postnatal, Laboratory, Pharmacy, Radiology and Physiotherapy.

In response to the COVID-19 pandemic, the hospital maintained one entrance gate as one of the preventive measures. Tents were constructed close to the entrance gate for COVID-19 screening. Individuals with signs and symptoms of COVID-19 or with a history of travel to the COVID-19 endemic areas were sent to COVID-19 tents for COVID-19 tests. Confirmed COVID-19 patients needing admission were admitted in three wards, and two of them had to be turned into COVID-19 wards. Tents were established as wards to create space for COVID-19 screening and admission of COVID-19 patients.

Management of a confirmed COVID-19 dead body including burial of it was only handled by a trained team of health professionals. The COVID-19 wards were managed by the existing staff at QECH. In response to the COVID-19 pandemic, QECH recruited temporary staff with Global Relief Aid (GRA) funding. HCWs with a confirmed COVID-19 test were isolated and were not allowed to work until they test COVID-19 negative. Personal Protective Equipment such as gloves, aprons and face mask were available for HCWs.

At the time of study HCWs at QECH had received AstraZeneca, and Johnson and Johnson vaccines. AstraZeneca is a two-dose vaccine, which is administered intramuscularly (0.5ml each) at 8 to 12 weeks while Johnson and Johnson is a single dose vaccine. To determine completeness of vaccines, HCWs had to receive two doses of AstraZeneca or one dose of Johnson and Johnson. In addition, confirmation of receipt of a vaccine was ascertained by viewing the HCWs COVID-19 vaccination card.

## Sample size

We used the Cochran's formula $n = \frac{Z^2 p(1-p)}{d^2}$, to calculate the sample size. As of June 2021, there were 1266 HCWs at QECH, which included 831 clinical professionals (doctors, clinicians, nurses, and physiotherapists), 128 non- clinical professionals (pharmacists, radiologists, and laboratory personnel), and 307 hospital attendants. We calculated the number of participants to be recruited from each stratified profession category by using the probability proportional to size formula. The prevalence proportion of vaccine uptake from a previous study conducted in Malawi in March 2021 after the COVID-19 vaccination rollout had commenced was 82.5%. Using a margin of error of 0.05 and a confidence interval of 95%, the minimum calculated sample size was 190. We estimated that there would be an attrition/non-response rate of 10%. Accounting for non-response, the final sample size was 209. The enrolment criteria included all consenting HCWs, aged $\geq$ 18 years with one-year working experience at QECH. We excluded HCWs aged less than 18 years and students.

## Data collection

The questionnaire was developed by the researcher. This was an interviewer-administered questionnaire and was informed by the study objectives and Behavioral and Social Drivers (BeSD) of the vaccination conceptual framework. It was then compared to other data collection tools of similar studies on the COVID-19 vaccine acceptance and uptake conducted in Malawi and other countries. The tools had similar perspectives on COVID-19 and COVID-19 vaccine. Some information on other tools was added to them. The data collection tools underwent expert review and were pretested to 10% of the calculated total sample size (19) of a population not included in the study. The data were collected electronically by the Principal Investigator (PI), using Open Data Kit (ODK), on a mobile phone. All interviews were conducted in English. The questionnaire consisted of sections on sociodemographic and economic information, knowledge, and perceptions of the COVID-19 and its vaccine.

Before collecting data, the eligibility criteria of the study were explained to departmental officer In-Charge. A simple random sampling technique was used to enroll participants. The study participants were recruited from the following wards/departments: the COVID-19 isolation units, COVID-19 ward, High Dependency Respiratory Unit (HDRU), Male and Female Medical wards (3B and 4A), Male and Female Surgical wards (5A and 5B), Accidents and Emergency department (A and E), AETC, Orthopaedic, Gynaecological, Chatinkha Nursery, Pediatric Moyo, Pediatric Special Care Ward (PSCW), Postnatal, Laboratory, Pharmacy, Radiology and Physiotherapy. Due to shortage of staff, six HCWs refused to be recruited into the study, because of work overload.

## Data analysis

Data was exported to Microsoft Excel spreadsheets. Data cleaning was done before data analysis, including editing the structural errors, and duplicates of participant identification numbers. Data were exported to Stata version 17.0 (StataCorp,Texas, USA) for analysis. We estimated the proportion of vaccine uptake with an exact 95% confidence interval. HCWs that were deemed fully vaccinated were those who received the first and second doses of the AstraZeneca COVID-19 vaccine or those who received other one-dose vaccines such as Johnson and Johnson. Continuous variables were summarized using means±standard deviations. The association between categorical variables was determined using Pearson's Chi-square test. A bivariate logistic regression model was used to identify the factors associated with the vaccine uptake. A multivariable logistic regression model was fitted to identify statistically significant factors in adjusted and unadjusted models. For the factor to be included in the model, it was nested for the likelihood ratio test to be valid. The independent variables were age, gender, marital status, education level, income level, work experience, medical insurance scheme, history of vaccine refusal, history of chronic illness, and occupation. The results were reported with 95% confidence intervals and p-values with significance level set at $p < 0.05$.

## Ethical consideration

The proposal of this study was submitted to the College of Medicine Research Ethics Committee (COMREC) P.09/21/3395 for scientific review and ethical approval. Permission to conduct the study at QECH was sought from the institution's Director before the study commenced. Participants provided written consent before participation in the study. The participants were fully informed of the study protocol and purpose. This permission was granted by signing the consent form. Participants were informed that they had a right to participate or to withdraw from the study. Names of participants were only available on the consent form; the participants were assigned a unique code for identification.

# Results

A total of 273 adults were enrolled in the study. The study enrolled more than the calculated sample size of 209, to improve the statistical power. From the 273 participants recruited in the study, 193 were fully vaccinated, representing 71% COVID-19 vaccine uptake. The mean age was 34 years (SD 9.2) and 61.9% were female. There were more clinical HCWs (61.9%), followed by hospital attendants (26.0%). Many HCWs reported attained tertiary education (76.6%), while only 4.8% had primary education. Most HCWs reported work experience of 0–3 years (57.9%) (Table 1).

## Bivariate associations

Age was significantly associated with the uptake of the vaccine (p = 0.02). The vaccine uptake increased among HCWs aged 30 years and above by 76.1% as compared to their counterparts aged 20–29 years (Table 2). Work experience of HCWs was associated with vaccine uptake with a p-value of 0.001 at 95% CI. HCWs with work experience of less than 3 years were less likely to get vaccinated as compared to their counterparts with work experience of 4 years and above. The ownership of the medical insurance scheme of HCWs was significantly associated with the uptake of the vaccine (p = 0.004). The vaccine uptake was higher among HCWs who were on medical insurance schemes with 84.9% than HCWs not on medical insurance scheme. History of vaccine refusal was statistically significant with a p-value of 0.002 at 95% CI. HCWs with no history of vaccine refusal had a higher chance of being vaccinated (73.4%) than their

**Table 1. Baseline socio-demographic characteristics of participants (N = 273).**

| Characteristic | Frequency | Percentage |
|---|---|---|
| **Age** | | |
| 20–29 | 114 | 41.8 |
| >30 | 159 | 58.2 |
| **Sex** | | |
| Female | 169 | 61.9 |
| **Marital status** | | |
| Married | 149 | 54.6 |
| Single | 102 | 37.4 |
| Divorced/widowed | 21 | 7.7 |
| Separation | 1 | 0.4 |
| **Religion** | | |
| Christian | 261 | 95.6 |
| Muslim | 9 | 3.3 |
| Other | 3 | 1.1 |
| **Profession** | | |
| Clinical HCWs | 169 | 61.9 |
| Non-clinical HCWs | 33 | 12.1 |
| Hospital attendants | 71 | 26 |
| **Working experience** | | |
| 0–3 years | 158 | 57.9 |
| 4 years and above | 115 | 42.1 |
| **Education** | | |
| Primary | 13 | 4.8 |
| Secondary | 51 | 18.7 |
| Tertiary | 209 | 76.6 |
| **Employer** | | |
| Government | 208 | 76.2 |
| NGO | 65 | 23.8 |
| **Household size** | | |
| One | 25 | 9.2 |
| Two-four | 132 | 48.4 |
| Five to seven | 104 | 38.1 |
| Eight and above | 12 | 4.4 |
| **Other sources of money** | | |
| No | 148 | 54.2 |
| Yes | 125 | 45.8 |
| **Medical Insurance Scheme** | | |
| Yes | 66 | 24.2 |
| **Chronic Illness** | | |
| Yes | 24 | 8.8 |
| **Ever refused vaccine** | | |
| Yes | 25 | 9.2 |

counterparts who ever refused vaccination. There was no statistical association between vaccine uptake and the following covariates: gender, profession/occupation, education level, household size, household income and history of chronic illness.

**Table 2. The association of HCWs and status of COVID-19 vaccination.**

| Characteristic | Vaccination Status | | | | P-value |
|---|---|---|---|---|---|
| | No | | Yes | | |
| | n | % | n | % | |
| **Age** | | | | | 0.02 |
| *20–29* | 42 | 36.8 | 72 | 63.2 | |
| *30 and above* | 38 | 23.9 | 121 | 76.1 | |
| **Sex** | | | | | 0.13 |
| *Male* | 25 | 24.0 | 79 | 76.0 | |
| *Female* | 55 | 32.5 | 114 | 67.5 | |
| **Marital status** | | | | | 0.50 |
| *Married* | 39 | 26.2 | 110 | 73.8 | |
| *Single* | 35 | 34.3 | 67 | 65.7 | |
| *Divorced/widowed* | 6 | 28.6 | 15 | 71.4 | |
| *Separation* | 0 | 0.0 | 1 | 100.0 | |
| **Religion** | | | | | 0.952 |
| *Christian* | 76 | 29.1 | 185 | 70.9 | |
| *Muslim* | 3 | 33.3 | 6 | 66.7 | |
| *Other* | 1 | 33.3 | 2 | 66.7 | |
| **Profession** | | | | | 0.45 |
| *Clinical HCWs* | 46 | 27.2 | 123 | 72.8 | |
| *Nonclinical HCWs* | 9 | 27.3 | 24 | 72.7 | |
| *Hospital attendants* | 25 | 35.2 | 46 | 64.8 | |
| **Working experience** | | | | | 0.00 |
| *0–3 Years* | 57 | 36.1 | 101 | 63.9 | |
| *4 years and above* | 23 | 20.0 | 92 | 80.0 | |
| **Education level** | | | | | 0.273 |
| *Primary* | 6 | 46.2 | 7 | 53.9 | |
| *Secondary* | 17 | 33.3 | 34 | 66.7 | |
| *Tertiary* | 57 | 27.3 | 152 | 72.7 | |
| **Employer** | | | | | |
| *Government* | 58 | 27.9 | 150 | 72.1 | |
| *NGO* | 22 | 33.9 | 43 | 66.2 | |
| **Household size** | | | | | 0.879 |
| *One* | 9 | 36.0 | 16 | 64.0 | |
| *Two-four* | 38 | 28.8 | 94 | 71.2 | |
| *Five to seven* | 30 | 28.9 | 74 | 71.2 | |
| *Eight and above* | 3 | 25.0 | 9 | 75.0 | |
| **Other sources of money** | | | | | 0.217 |
| *No* | 48 | 32.4 | 100 | 67.6 | |
| *Yes* | 32 | 25.6 | 93 | 74.4 | |
| **Medical Insurance Scheme** | | | | | 0.004 |
| *No* | 70 | 33.8 | 137 | 66.2 | |
| *Yes* | 10 | 15.2 | 56 | 84.9 | |
| **Chronic illness** | | | | | 0.34 |
| *No* | 75 | 30.1 | 174 | 69.9 | |
| *Yes* | 5 | 20.8 | 19 | 79.2 | |
| **History of vaccine refusal** | | | | | 0.002 |
| *No* | 66 | 26.6 | 182 | 73.4 | |

*(Continued)*

**Table 2.** (Continued)

| Characteristic | Vaccination Status | | | | P-value |
|---|---|---|---|---|---|
| | No | | Yes | | |
| | n | % | n | % | |
| *Yes* | 14 | 56.0 | 11 | 44.0 | |

N = 273

## Multivariable models

Factors associated with the vaccine uptake in crude and adjusted odds ratio are shown in Table 3. Being female (AOR: 0.54, 95% CI: 0.30–0.96) was statistically associated with the uptake of the vaccine, females were 0.5 times less likely to get vaccinated against COVID-19.

Ownership of medical insurance schemes (AOR: 2.90, 95% CI: 1.36–6.18) was statistically associated with the vaccine uptake. HCWs on medical insurance cover were 2.9 times more likely to get vaccinated. History of vaccine refusal (AOR: 0.29, 95% CI: 0.12–0.70) was also statistically associated with the vaccine uptake. HCWs who had refused a vaccine before were 0.2 times less likely to get vaccinated.

## Discussion

The main findings of this study that assessed the uptake of the COVID-19 vaccine and associated factors among HCWs at QECH in Blantyre show that out of the 273 HCWs enrolled into the study 71% were fully vaccinated. Being female (AOR: 0.54, 95% CI: 0.30–0.96), ownership of medical insurance schemes (AOR: 2.90, 95% CI: 1.36–6.18) and history of vaccine refusal (AOR: 0.29, 95% CI: 0.12–0.70) were significantly related to vaccine uptake. Age and work experience were statistically significant. The level of education, the income of HCWs and occupation did not determine whether HCWs got vaccinated against COVID-19.

This study found a relatively high vaccine uptake among HCWs during the active roll out of national COVID-19 vaccination program. A previous study conducted in Malawi during the commencement of national COVID-19 vaccination program found a slightly higher COVID-19 vaccine uptake of 82.5% [13]. The decline in the vaccine uptake could be attributed to the facts that the previous study was conducted before a decline of the cases when the

**Table 3. Crude and adjusted analysis of factors associated with the COVID-19 vaccine.**

| Characteristic | Crude OR | 95% CI | p-value | Adjusted OR | 95% CI | p-value |
|---|---|---|---|---|---|---|
| **Age** | | | | | | |
| *20–29* | 1 | | | 1 | | |
| *>30* | 1.86 | 1.09, 3.14 | 0.021 | 1.55 | 0.88, 2.69 | 0.123 |
| **Sex** | | | | | | |
| *Male* | 1 | | | 1 | | |
| *Female* | 0.66 | 0.37, 1.14 | 0.135 | 0.54 | 0.30, 0.96 | 0.039 |
| **Medical Insurance Scheme** | | | | | | |
| *No* | 1 | | | 1 | | |
| *Yes* | 2.86 | 1.37, 5.94 | 0.005 | 2.90 | 1.36, 6.18 | 0.006 |
| **History of vaccine refusal** | | | | | | |
| *No* | 1 | | | 1 | | |
| *Yes* | 0.28 | 0.12, 0.65 | 0.003 | 0.29 | 0.12, 0.70 | 0.006 |

second dose of AstraZeneca was being rolled-out [13]. The studies were conducted during second and fourth COVID-19 waves respectively, and it was when there were many deaths [14]. However, the vaccine uptake in this study is much higher compared to the results reported from Nigeria and Somalia, which estimated the prevalence to be at 33% and 37.4% respectively. The main constraint of vaccination among HCWs in Nigeria was limited vaccines, while in, Somalia the uptake of the vaccine was affected by a lack of security and accessibility challenges [15, 16].

This study found that age is significantly associated with vaccine uptake. Those 30 years or older, were more likely to be vaccinated against COVID-19. This infers that people are more likely to accept the vaccination as they age. This result remains consistent with previous findings in developed and developing countries that reported higher vaccine acceptance and uptake among the old population than in the young population [5, 13, 17, 18]. Another explanation is that older age is a known risk factor for COVID-19 complications which increases the perceived vulnerability in this age group which could have triggered the higher uptake of the vaccine [19]. Furthermore, it is coherent to BeSD of vaccination framework which states that individual's perceived risk towards a disease influences their vaccine acceptance and uptake [12]. The lower vaccine uptake among young adults can be attributed to their easy access to social media where they are likely to come across conflicting information, myths, conceptions, and conspiracy theories on vaccination which may eventually result in low vaccine uptake [20].

Contrary to the findings of this study, similar studies conducted in Ghana and the Kingdom of Saudi Arabia reported higher vaccine acceptance and uptake among the younger population who were not vulnerable to COVID-19 complications than the older population [21, 22]. This difference could be explained by the myths about the vaccine, and the study sample size with a large proportion of young adults of 18–29 years respectively [20, 22]. We suggest that policy makers and local public health authorities should target subgroups, such as young adults for COVID-19 vaccination campaigns, to optimise the vaccine uptake [20]. The campaign should also share them on information about the importance of the vaccine and dispel the myths.

This study demonstrated a strong association between gender and vaccine uptake. Being male increased the likelihood of accepting the vaccine. This finding is like the results reported from several countries which revealed low vaccine uptake among females as compared to males [23, 24]. This finding could be attributed to fear of vaccine safety and efficacy, inadequate knowledge, and misconceptions that the vaccine may cause infertility in females [17, 23, 25, 26]. Initially, the guidelines excluded pregnant women and those who intended to get pregnant from vaccinating [27]. The lower uptake among females could also be due to conflicting information regarding the safety of the vaccine in pregnant and lactating women [28]. In addition, this result could be due to high COVID-19 individual perceived risk among males than in females, as men are at increased risk of contracting COVID-19 due to biological make-up [23, 29]. Furthermore, this result could also be due to high mortality and morbidity rates among males than females [30].

The level of education showed no significant association with vaccine uptake in this study. However, the uptake of the vaccine has been observed to be relatively high among HCWs with a tertiary level of education, than in HCWs with a secondary and primary level of education. Some high-income countries such as the USA, Portugal and Austria revealed that individuals with a university degree are more likely to get vaccinated since they trust that the vaccines are safe and effective than their counterparts without a university degree [7, 31, 32]. In contrast, a study in Kenya found that due to inadequate and conflicting information on social media about the vaccine safety and efficacy, youths in the general population with post-secondary education were less likely to get vaccinated against COVID-19, as compared to their counter

parts with primary and secondary education [33]. In Malawi, a study found no significant association between vaccine uptake and the level of education among HCWs [13]. What is most notable is the conflicting information about vaccines on social media and other social platforms has caused suboptimal uptake of the vaccine, and other vaccines even among health-care professionals [17, 34–36].

This study did not establish any significant association between the rank of profession/ occupation and vaccine uptake. This is contrary to what several studies in low, middle, and high-income countries found. The studies found that low-cadre HCWs are less likely to accept the vaccine since many of them are not in close contact with patients, compared to high-cadre HCWs [13, 17, 37]. High vaccine hesitancy has been observed in HCWs not in direct patient care, most of them opted to wait and review data on the safety and efficacy of the vaccine before deciding to get vaccinated [17]. However, in Malawi, due to understaffing in most public hospitals, the low-cadres HCWs are highly involved in clinical duties and are in close contact with patients; hence they too are at increased risk of contracting COVID-19 [38].

This study also found that vaccine uptake increased as the work experience of HCWs increased. This finding is consistent with results reported from the USA, which demonstrated an increase in vaccine uptake among HCWs with more than ten years of work experience, as compared to their counterparts with less than ten years of work experience [19]. Mostly, the more experienced HCWs tend to be older, which puts them at a high risk of various chronic diseases, hence the willingness to get vaccinated against COVID-19. Furthermore, their understanding of outbreaks and vaccines overtime optimises the COVID-19 vaccine uptake as compared to their counter parts with less work experience [17].

The history of vaccine refusal was significantly associated with the uptake of the vaccine. Among HCWs who never refused any vaccine before, 26.6% of them did not get vaccinated against COVID-19. Similarly, a study in the USA, found that parents who were not hesitant to childhood immunisations were reluctant to vaccinate their children and themselves against the COVID-19. This finding could be attributed to persisting negative information and conspiracy theories about the vaccine on social media and other social platforms [25, 39]. We suggest that governments should build trust in people about the vaccine using mass communication to the public to address the vaccine conspiracy theories [13, 25, 39].

There was also a significant association between vaccine uptake and ownership of medical insurance schemes. High uptake was observed among HCWs on medical insurance cover (p<0.004) as compared to HCWs not on any medical insurance cover. This finding correlates with results reported from the USA which demonstrated a higher vaccine uptake among individuals on medical insurance schemes than those not on any medical insurance schemes. However, this finding contradicts results reported from another study in the USA and Ethiopia which found no significant association between vaccine uptake and ownership of medical insurance schemes. Studies conducted in Kenya, Gabon, the Democratic Republic of Congo (DRC), Cameroon, and Burkina Faso revealed that an increase in age, education level and household wealth status showed a consistently positive relationship with ownership of medical insurance schemes. Hence HCWs with ownership of medical insurance schemes were likely to be of a higher social-economic group than their counterparts, not on medical insurance schemes [39]. This finding could be because the study recruited NGOs and government employed HCWs. Hence, the NGO-based HCWs were more likely to own health medical insurance cover than HCWs working in the government as per their work conditions respectively [40]. Furthermore, HCWs in medical insurance schemes had better access to information on the vaccine through private health firms, hence, the higher vaccine uptake as compared to HCWs, not in medical insurance schemes [41]. In addition, the finding in this

study is likely due to the smaller sample size of HCWs on medical insurance cover. This finding needs further investigation.

This study has demonstrated no significant association between income level and vaccine uptake. This finding differs from results reported from the USA, and a systematic review, which found that the vaccine uptake increased among individuals with a high level of income than their counterparts with low-income levels [23, 42, 43]. Since the level of education and occupation are social-economic factors, and they showed no significant association vaccine uptake, the income level had shown no significant association. This finding reveals that during the COVID-19 pandemic, the income level of individuals did not significantly influence vaccine uptake. The results could be explained by the fact that the study participants were within same income levels hence there was less heterogeneity.

## Study strengths and limitation

The study was conducted during the inception of the COVID-19 vaccination roll-out; hence it demonstrated the actual vaccine uptake, the associated factors, and other correlates among HCWs at QECH.

However, the study design was a quantitative cross-sectional study, which does not allow the exploration of causal relationships. Therefore, a mixed-method study design is recommended so that the qualitative component gives an understanding of factors that optimise uptake of the COVID-19 vaccine. It was a single site study, hence study results should be generalized with caution. We recommend a similar study in tertiary hospitals in central and northern region of Malawi.

## Conclusion

In this study of assessing the uptake of COVID-19 vaccine and identifying the associated factors among HCWs in Malawi, there was high vaccine uptake among HCWs at QECH. Some risk factors for severe COVID-19 such as advancing in age and work experience are persuasive to vaccination. We recommend intensified COVID-19 vaccination campaigns for subgroups such as young adults and female HCWs.

## Supporting information

**S1 Questionnaire. Participant's questionnaire.**
(DOCX)

## Acknowledgments

This manuscript is part of the "COVID-19 vaccine and associated factors among health care workers at a tertiary hospital in Malawi". The authors are grateful to the study participants for their voluntary participation and the Director of Health and Social Services for Queen Elizabeth Central Hospital for institutional support.

## Author Contributions

**Conceptualization:** Angella Joseph Chikhoza.

**Data curation:** Angella Joseph Chikhoza.

**Formal analysis:** Angella Joseph Chikhoza, Wingston Ng'ambi, Alinane Linda Nyondo-Mipando.

**Investigation:** Angella Joseph Chikhoza, Wingston Ng'ambi, Alinane Linda Nyondo-Mipando.

**Methodology:** Angella Joseph Chikhoza, Wingston Ng'ambi, Alinane Linda Nyondo-Mipando.

**Resources:** Angella Joseph Chikhoza.

**Supervision:** Wingston Ng'ambi, Alinane Linda Nyondo-Mipando.

**Validation:** Wingston Ng'ambi, Alinane Linda Nyondo-Mipando.

**Writing – original draft:** Angella Joseph Chikhoza.

**Writing – review & editing:** Wingston Ng'ambi, Alinane Linda Nyondo-Mipando.

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
