## [Decision Letter · Decision Letter 0]

22 May 2024

PGPH-D-24-00083

Uptake of COVID-19 Vaccine and Associated Factors among Healthcare Workers at Queen Elizabeth Central Hospital in Malawi: A Cross-Section Study

Dear Dr. Chikhoza,

Thank you for submitting your manuscript to PLOS Global Public Health. After careful consideration, we feel that it has merit but does not fully meet PLOS Global Public Health’s publication criteria as it currently stands. Therefore, we invite you to submit a revised version of the manuscript that addresses the points raised during the review process.

We look forward to receiving your revised manuscript.

Kind regards,

Edina Amponsah-Dacosta, Ph.D., MPH

Academic Editor

Journal Requirements:

Additional Editor Comments (if provided):

Reviewers' comments:

Reviewer's Responses to Questions

**Comments to the Author**

1. Does this manuscript meet PLOS Global Public Health’s publication criteria? Is the manuscript technically sound, and do the data support the conclusions? The manuscript must describe methodologically and ethically rigorous research with conclusions that are appropriately drawn based on the data presented.

Reviewer #1: Partly

Reviewer #2: Yes

2. Has the statistical analysis been performed appropriately and rigorously?

Reviewer #1: I don't know

Reviewer #2: Yes

3. Have the authors made all data underlying the findings in their manuscript fully available (please refer to the Data Availability Statement at the start of the manuscript PDF file)?

Reviewer #1: Yes

Reviewer #2: Yes

4. Is the manuscript presented in an intelligible fashion and written in standard English?

Reviewer #1: No

Reviewer #2: Yes

5. Review Comments to the Author

Reviewer #1: Introduction:

SARS-CoV-2 stands for Severe acute respiratory syndrome coronavirus 2 and not Severe Acute Respiratory Syndrome (SARS-CoV-2) coronavirus.

The sentence ‘Vaccination against SARS-CoV-2 promises to end the COVID-19 pandemic’ as the WHO declared the end of pandemic phase of COVID-19 in May 2023.

Add the latest mortality data (as of April 2024) and not as of 17 August 2023.

Method:

The study objectives are premised on HCW COVID-19 vaccine and associated factors. However, the population considered is of health professionals (1266) and not of HCWs (1710).

Please recheck the minimum sample size calculation. Using a margin of error of 0.05 and a confidence interval of 95%, the minimum calculated sample size of a population of 1266 cannot be 222.

More details of the random sampling technique should be provided.

Clear inclusion and exclusion criteria need to be provided.

Were study participants recruited from all wards/Departments? This is not made clear (although a list of wards/departments is given).

How did you ensure validity and reliability of the research process and the data collected.

Results:

Report results to the same number of decimal places.

Report first on the overall vaccine uptake.

The information provided in Table 1

Table 2 needs to be reworked and reformatted.

Table 2 – Add value for N

Table 2 - What is ‘Other’ marital status?

Table 2 – Segregate the age ranges further. Two categories is unusual and does not paint a full picture of the vaccine uptake by age range

Table 2 – Segregate the professions further. E.g. Medical doctors, professional nurses, laboratory technicians, pharmacists, dentists, physiotherapists etc

Table 3 – Adjusted OR and p-values for some variables are missing

Study strengths and Limitations

The statement, ‘The sample size was powered for high vaccine uptake, but as results show it is on the contrary hence this is a limitation that needs further investigation’, needs to be rephrased. It is not clear.

Reviewer #2: Overall this is good work highlighting a pertinent public health problem of COVID 19 vaccine uptake and associated determinants amidst high levels of vaccines hesitancy. I am impressed with how the methodology are rigorous, and clear. This is commendable, however the authors need to provide more information on the following

1. In the introduction, it would be important if the source of the following data is recognized or cited; In Malawi, 1061 HCWs in the Blantyre district tested SARS-CoV-2 antibodies positive as of August 2023.

2. The author has put up a strong discussion, however i have noted that in the last statement of the second paragraph, the author is issuing what i believe to be a limitation, and recommendation, as demonstrated in the statement below (The

study results should be generalized with caution since it was a single site study. We recommend

a similar study in tertiary hospitals in central and northern region of Malawi). I believe the authors should focus on the discussion, and add such on the study limitations and recommendation.

3. With regards to the age brackets given, those less than 30 and those above thirty, even though advanced age is a risk factor severe COVID-19 disease, the authors need to clearly define which advanced age, as 30-40 is also considered relatively young, perhaps the authors can provide more details as how many were above the age of 40.

6. PLOS authors have the option to publish the peer review history of their article (what does this mean?). If published, this will include your full peer review and any attached files.

**Do you want your identity to be public for this peer review?** For information about this choice, including consent withdrawal, please see our Privacy Policy.

Reviewer #1: No

Reviewer #2: **Yes: **Adriano Focus Lubanga

---

## [Editor Report · Decision Letter 1]

26 Aug 2024

PGPH-D-24-00083R1

Uptake of COVID-19 Vaccine and Associated Factors among Healthcare Workers at Queen Elizabeth Central Hospital in Malawi: A Cross-Section Study

Dear Dr. Chikhoza,

Thank you for submitting your revised manuscript to PLOS Global Public Health. I am returning this submission to you as I have noted that you did not address Reviewer #2's comments or provide a detailed rebuttal to the issues raised. I have communicated this in an earlier email to the corresponding author, providing the comments referred to once again for reference. Please note that this is a final opportunity to address all comments raised and refine your manuscript for consideration by the peer-reviewers.

We look forward to receiving your revised manuscript.

Kind regards,

Edina Amponsah-Dacosta, Ph.D., MPH

Academic Editor

Journal Requirements:

Additional Editor Comments (if provided):

Reviewers' comments:

Reviewer #2: Overall this is good work highlighting a pertinent public health problem of COVID 19 vaccine uptake and associated determinants amidst high levels of vaccines hesitancy. I am impressed with how the methodology are rigorous, and clear. This is commendable, however the authors need to provide more information on the following

1. In the introduction, it would be important if the source of the following data is recognized or cited; In Malawi, 1061 HCWs in the Blantyre district tested SARS-CoV-2 antibodies positive as of August 2023.

2. The author has put up a strong discussion, however i have noted that in the last statement of the second paragraph, the author is issuing what i believe to be a limitation, and recommendation, as demonstrated in the statement below (The study results should be generalized with caution since it was a single site study. We recommend a similar study in tertiary hospitals in central and northern region of Malawi). I believe the authors should focus on the discussion, and add such on the study limitations and recommendation.

3. With regards to the age brackets given, those less than 30 and those above thirty, even though advanced age is a risk factor severe COVID-19 disease, the authors need to clearly define which advanced age, as 30-40 is also considered relatively young, perhaps the authors can provide more details as how many were above the age of 40.

---

## [Decision Letter · Decision Letter 2]

8 Oct 2024

PGPH-D-24-00083R2

Uptake of COVID-19 Vaccine and Associated Factors among Healthcare Workers at Queen Elizabeth Central Hospital in Malawi: A Cross-Section Study

Dear Dr. Chikhoza,

Thank you for submitting your manuscript to PLOS Global Public Health. After careful consideration, we feel that it has merit but does not fully meet PLOS Global Public Health’s publication criteria as it currently stands. Therefore, we invite you to submit a revised version of the manuscript that addresses the points raised during the review process.

EDITOR: Kindly carefully review the manuscript and ensure that all grammatical and typographical errors are appropriately addressed. The Reviewer has raised some but not all errors. Please give appropriate attention to this to ensure that the core meaning of the manuscript is not lost.

We look forward to receiving your revised manuscript.

Kind regards,

Edina Amponsah-Dacosta, Ph.D., MPH

Academic Editor

Journal Requirements:

Additional Editor Comments (if provided):

Reviewers' comments:

Reviewer's Responses to Questions

**Comments to the Author**

1. If the authors have adequately addressed your comments raised in a previous round of review and you feel that this manuscript is now acceptable for publication, you may indicate that here to bypass the “Comments to the Author” section, enter your conflict of interest statement in the “Confidential to Editor” section, and submit your "Accept" recommendation.

Reviewer #1: (No Response)

Reviewer #2: All comments have been addressed

2. Does this manuscript meet PLOS Global Public Health’s publication criteria? Is the manuscript technically sound, and do the data support the conclusions? The manuscript must describe methodologically and ethically rigorous research with conclusions that are appropriately drawn based on the data presented.

Reviewer #1: Yes

Reviewer #2: Yes

3. Has the statistical analysis been performed appropriately and rigorously?

Reviewer #1: Yes

Reviewer #2: Yes

4. Have the authors made all data underlying the findings in their manuscript fully available (please refer to the Data Availability Statement at the start of the manuscript PDF file)?

Reviewer #1: Yes

Reviewer #2: Yes

5. Is the manuscript presented in an intelligible fashion and written in standard English?

Reviewer #1: Yes

Reviewer #2: Yes

6. Review Comments to the Author

Reviewer #1: Results:

• Report results in the table and in-text to the same number of decimal places so that there is consistency throughout the document.

• Table 2 – n=1 for ‘Other’ marital status. You have defined Other’ marital status as, separation, cohabitation and registered partnerships. Why not specify especially as only one person satisfied this criteria

• Table 3 - Fix typographical errors (e.g. MedicalInsuranceScheme and Historyofvaccinerefusal)

Reviewer #2: Dear authors

Thank you for responding to our previous comments. I don't have issues with the paper any more.

7. PLOS authors have the option to publish the peer review history of their article (what does this mean?). If published, this will include your full peer review and any attached files.

**Do you want your identity to be public for this peer review?** For information about this choice, including consent withdrawal, please see our Privacy Policy.

Reviewer #1: No

Reviewer #2: No

---

## [Editor Report · Decision Letter 3]

8 Nov 2024

Uptake of COVID-19 Vaccine and Associated Factors among Healthcare Workers at Queen Elizabeth Central Hospital in Malawi: A Cross-Section Study

PGPH-D-24-00083R3

Dear Mrs Chikhoza,

We are pleased to inform you that your manuscript 'Uptake of COVID-19 Vaccine and Associated Factors among Healthcare Workers at Queen Elizabeth Central Hospital in Malawi: A Cross-Section Study' has been provisionally accepted for publication in PLOS Global Public Health.

Best regards,

Edina Amponsah-Dacosta, Ph.D., MPH

Academic Editor